

# Characterization of gut microbiota in adults with coronary atherosclerosis

Yu Dong[1,2,*], Rui Xu[1,2,*], Xiaowei Chen[1,2], Chuanli Yang[1,2], Fei Jiang[1,2], Yan Shen[1,2], Qiong Li[1,2], Fujin Fang[1,2], Yongjun Li[3] and Xiaobing Shen[1,2]

[1] Key Laboratory of Environmental Medicine Engineering, Ministry of Education, School of Public Health, Southeast University, Nanjing, China

[2] Department of Epidemiology and Health Statistics, School of Public Health, Southeast University, Nanjing, China

[3] Department of Cardiology, Zhongda Hospital, Southeast University, Nanjing, China

[*] These authors contributed equally to this work.

## ABSTRACT

**Background**. Cardiovascular disease, which is mainly caused by coronary atherosclerosis, is one of the leading causes of death and disability worldwide. Gut microbiota likely play an important role in coronary atherosclerosis. This study aims to investigate the microbiota profile of adults with coronary atherosclerosis to provide a theoretical basis for future research.

**Methods**. Fecal samples were collected from 35 adult patients diagnosed with coronary atherosclerosis and 32 healthy adults in Nanjing, China, and the V3-V4 region of 16S rDNA genes was sequenced using high-throughput sequencing. Differences in alpha diversity, beta diversity, and gut microbiota composition between the two groups were then compared.

**Results**. A beta diversity analysis revealed significant differences between adults with coronary atherosclerosis and controls, but there was no statistical difference in alpha diversity between the two groups. There were also differences in the composition of the gut microbiota between the two groups. The genera, *Megamonas*, *Streptococcus*, *Veillonella*, *Ruminococcus_torques_group*, *Prevotella_2*, *Tyzzerella_4*, were identified as potential biomarkers for coronary atherosclerosis.

**Conclusion**. There are some differences in the gut microbiota of adults with coronary atherosclerosis compared to healthy adults. The insights from this study could be used to explore microbiome-based mechanisms for coronary atherosclerosis.

Corresponding authors
Yongjun Li, 2650659538@qq.com
Xiaobing Shen, xb.shen@seu.edu.cn

## INTRODUCTION

Cardiovascular diseases are associated with high rates of morbidity and mortality as well as immense and increasing healthcare costs (*Benjamin et al., 2018*; *World Health Organization, 2020*). Atherosclerosis, the leading cause of heart disease and stroke, accounts for approximately 50% of all cardiovascular deaths (*Sanchez-Rodriguez et al., 2020*). The key processes responsible for the development of atherosclerosis are the formation and accumulation of foam cells within the lipid-rich subendothelial space of the affected artery; macrophages engulf a large amount of lipoproteins under the endothelium to form foam

cells, then accumulate as a characteristic 'fatty streak,' leading to a greater likelihood of plaque rupture and blood vessel occlusion (*Maguire, Pearce & Xiao, 2019*). Endothelial dysfunction, chronic inflammation, hyperglycemia, and oxidative stress can all affect the development of atherosclerosis (*Katakami, 2018*). Atherosclerosis happens over time and is associated with long-term and accumulative exposure to the causal changeable risk factors.

The gut microbiome is one of the richest microbial environments in the human body. It is estimated that the number of human gut microbiota is more than 10 times the number of human cells. Although there have been significant advances in this research area, the complexity of this ecosystem means there is still much that is unknown. Alterations in bacterial composition and the metabolic compounds produced by these bacteria have been associated with the pathogenesis of many metabolic and inflammatory diseases, such as obesity (*Crovesy, Masterson & Rosado, 2020*), type 2 diabetes (*Yang et al., 2021*), nonalcoholic fatty liver disease (*Kolodziejczyk et al., 2019*), inflammatory bowel disease (*Khan et al., 2019*), and rheumatic disease (*Zhong et al., 2018*). Intestinal bacteria play an important role in regulating metabolism and inflammation in the body. There may also be a close association between intestinal bacteria and the development of coronary atherosclerosis because atherosclerosis is both a metabolic disorder and a chronic inflammatory disorder (*Nazarian-Samani, Sewell & Rafieian-Kopaei, 2020*; *Poznyak et al., 2022*).

Accumulating evidence has linked gut microbiota to coronary atherosclerosis. *Tuomisto et al. (2019)*, in an autopsy study of male patients, found several bacteria that were present in both the coronary plaque and the feces, indicating that bacteria present in the plaque could be of intestinal origin. *Nakajima et al. (2022)* found several kinds of intestinal bacteria that were related to vulnerable plaque features in patients with cardiovascular diseases. Another study found that mice gavaged with feces from myocardial infarction patients developed intensive arterial stiffness (*Liu et al., 2020*). These studies all indicate a close relationship between intestinal bacteria and atherosclerosis. However, only a few studies have examined intestinal bacteria in Chinese adults with coronary atherosclerosis (*Hu et al., 2021*; *Jie et al., 2017*). The identification of a disease-associated gut microbial composition for coronary atherosclerosis is the first step in developing related strategies for early diagnosis and treatment. In this study, we focused on identifying the distinctive profile of gut microbiota in Chinese adults with coronary atherosclerosis through 16S rDNA sequencing in fecal samples.

## MATERIALS AND METHODS

### Sample collection

This study was approved by the Clinical Research Ethical Committee of Zhongda Hospital Affiliated with Southeast University (Grant No. 2021ZDSYLL147-P01). All participants voluntarily agreed to participate in the study and signed informed consent forms.

Adults with coronary atherosclerosis were enrolled in the study if they exhibited ≥ 30% stenosis in at least one coronary artery indicated by coronary angiography; healthy adults were enrolled in the study as controls if they were free of atherosclerosis by medical

examination. Subjects were excluded if they had gastrointestinal dysfunction, recent diarrhea, a history of ulcerative colitis, Crohn's disease or other gastrointestinal diseases, diabetes, tumors, or had taken antibiotics, probiotics, or herbal medicines within one month prior to fecal sample collection.

A total of 67 participants were recruited from Zhongda Hospital between September 2021 and December 2021: 35 participants in the coronary atherosclerosis group and 32 individuals in the control group. Fecal samples were collected from the participants between 8 and 9 am and stored in sterile fecal collection tubes, which were then placed in ice boxes and immediately transferred to a $-80\ ^{\circ}$C refrigerator for storage.

### DNA extraction, PCR amplification, and 16S rDNA gene sequencing

Microbial DNA was extracted from the feces using a stool DNA kit (Tiangen Biotech, Beijing, China) using the steps outlined by the manufacturer. The V3-V4 region of 16S rDNA genes was amplified by PCR with the 338F (5′-ACTCCTACGGGAGGCAGCA-3′) and 806R (5′-GGACTACHVGGGTWTCTAAT-3′) primers (*Caporaso et al., 2011*; *Huws et al., 2007*). The reaction system consisted of 50 ng of template DNA, 0.3 µl of each primer (10 µM), 2 µl of dNTPs (2 mM each), 5 µl of KOD FX Neo Buffer, 0.2 µl of KOD FX Neo, and double distilled $H_2O$, for a total volume of 10 µl (TOYOBO, Osaka, Japan). The amplification procedure was as follows: initial denaturation at 95 °C for 5 min, followed by denaturation at 95 °C for 30 s, annealing at 50 °C for 30 s, and extension at 72 °C for 40 s. This procedure was repeated for 25 cycles, followed by a final extension at 72 °C for 7 min.

The PCR products were then amplified by the second-round tailed PCR to add indices and adapter sequences. In the second round of PCR amplification, we used 10 µl of $2 \times Q5^{\circledR}$ High-Fidelity Master Mix, 2.5 µl 2 µM of each primer (MPPI-a and MPPI-b), and 5 µl of the first-round amplification product. This second round of amplification was simpler than the first and included: initial 30 s denaturation at 98 °C, 10 cycles of denaturation at 98 °C for 10 s, annealing at 65 °C for 30 s, an extension at 72 °C for 30 s, and finally an extension at 72 °C for 5 min.

The amplified products were purified and recovered using 1.8% agarose gel electrophoresis. The sequencing was performed using Illumina's Novaseq 6000 platform with Biomarker Technologies Co., Ltd. (Beijing, China) providing the sequencing service.

### Sequence data processing

The raw reads obtained from sequencing were filtered using Trimmomatic version 0.33 software (*Bolger, Lohse & Usadel, 2014*), then the primer sequences were identified and removed using Cutadapt version 1.9.1 to obtain clean reads without primer sequences (*Martin, 2011*). The clean reads from each sample were then spliced with a minimum overlap length of 10bp, minimum similarity within overlapping region of 90%, and a maximum mismatch of 5 bp using USEARCH version 10.0 (*Edgar, 2013*). Next, the chimeric sequences were identified and removed to obtain the final effective reads using UCHIME version 4.2 (*Edgar et al., 2011*). The high-quality reads generated from these steps were then used in the following analysis.

**Table 1 Age and gender distribution of case and control groups.**

| Variables | Case | Control | *P* value |
|---|---|---|---|
| Age (mean ±SD, years) | 59.7 ± 8.4 | 58.6 ± 11.6 | 0.667 |
| Gender | | | |
| Male | 19 (54.3%) | 16 (50%) | 0.726 |
| Female | 16 (45.7%) | 16 (50%) | |
| Total | 35 | 32 | |

## OTU cluster and species annotation

The qualified sequences with more than 97% similarity thresholds were allocated to one operational taxonomic unit (OTU) using USEARCH (version 10.0) (*Edgar, 2013*). The taxonomy annotation of the OTUs was performed based on the naive Bayes classifier in QIIME2 using the SILVA database (release 138.1) with a confidence threshold of 70% (*Quast et al., 2013*).

## Diversity and LEfSe analysis

Alpha diversity, reflecting the species richness of individual sample and the species diversity, was evaluated in this study by the ACE, Chao1, Simpson, and Shannon indices. Beta diversity, which measures the difference in community composition, was evaluated using the unweighted uniFrac distance matrix, and a principal coordinate analysis (PCoA) of unweighted uniFrac distance was plotted. An analysis of similarities (Anosim) was performed to test the beta diversity separations. Potential microbial biomarkers were identified through a linear discriminant analysis effect size (LEfSe) analysis with an alpha value of 0.05 and an effect size threshold of 2. These analyses were performed using the BMK Cloud (http://www.biocloud.net).

## Statistical analysis

Differences in the categorical variables between the two groups were determined using the chi-square test. Differences in the numerical variables conforming to normal distribution and equal variance between the coronary atherosclerosis and control samples were assessed using the *t* test, and the results were displayed as mean ± standard deviation (SD). The data were analyzed using SPSS 25.0 software. $P < 0.05$ was considered statistically significant.

# RESULTS

## Study participants

Overall, 67 samples were collected from 35 adults with coronary atherosclerosis and 32 healthy controls. There were no significant differences in age or sex between the two groups ($P > 0.05$; Table 1).

## Microbial diversity

A total of 4,866,283 effective reads were generated. An average of 72,631 effective reads were generated per sample with 45,914 effective reads being the smallest number generated from a single sample (Table 2).

**Table 2  Diversity evaluation of gut microbiota in two groups.**

| Group | Effective reads mean, min ∼max | Coverage mean, min ∼max | OTU mean ± SD | ACE mean ± SD | Chao1 mean ± SD | Simpson mean ± SD | Shannon mean ± SD |
|---|---|---|---|---|---|---|---|
| Case | 72799, 45914∼105299 | 0.9994, 0.9988∼0.9996 | 244.8 ± 30.6 | 279.8 ± 28.3 | 282.1 ± 29.0 | 0.9 ± 0.1 | 4.5 ± 0.7 |
| Control | 72448, 55904∼76791 | 0.9994, 0.9991∼0.9996 | 228.3 ± 28.7 | 275.9 ± 30.3 | 275.3 ± 34.3 | 0.9 ± 0.1 | 4.5 ± 0.6 |

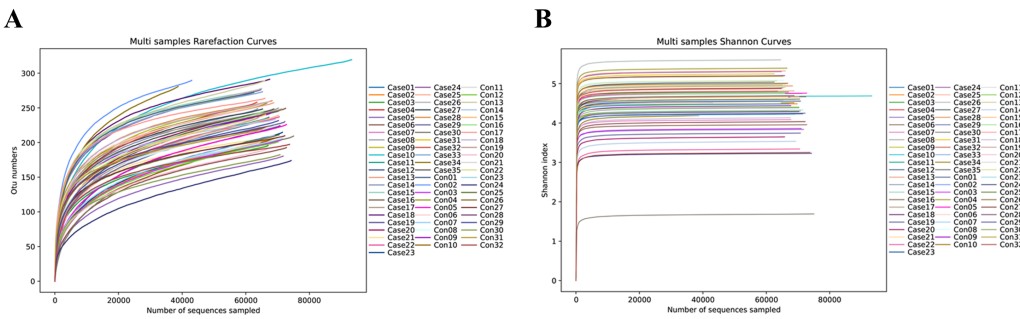

**Figure 1  Rarefaction curve and Shannon curve.** (A) the rarefaction curve of each sample; (B) the Shannon curve of each sample. Case, adult with coronary atherosclerosis; Con, healthy control.

The rarefaction curve (Fig. 1A) and Shannon index curve (Fig. 1B) were flat, indicating that the sequencing depth was sufficient to detect the majority of microbial species in the samples.

The results of the $t$ test on the OTU counts of the two groups showed that the coronary atherosclerosis group contained a greater number of OTUs than the control group ($P = 0.027$; Table 2).

The differences in each index of alpha diversity between the two groups are shown below in Table 2. There were no significant differences in the ACE, Chao1, Simpson, and Shannon indices observed by the $t$ test.

The PCoA based on the abundance of OTUs revealed differences in the microbial composition of the coronary atherosclerosis group and the healthy control group (Fig. 2A), and the Anosim results indicated significant differences in gut microbiota between the two groups (Fig. 2B).

## Dominant microbiome in fecal samples

At the phylum level, the top 10 phyla were: *Firmicutes, Bacteroidetes, Proteobacteria, Actinobacteria, Fusobacteria, Verrucomicrobia, Tenericutes, Patescibacteria, Cyanobacteria,* and *Synergistetes* (Fig. 3A). An increase in the abundance of *Firmicutes* (56.25% *vs* 49.93%, $P = 0.070$) and a decrease in the abundance of *Bacteroidetes* were observed in the coronary atherosclerosis group (33.98% *vs* 39.59%, $P = 0.061$), but these differences did not reach statistical significance. Among the top 10 phyla, the phyla with significant differences between the two groups were *Cyanobacteria* and *Patescibacteria*.

At the family level, the top 10 families were: *Bacteroidaceae, Ruminococcaceae, Lachnospiraceae, Veillonellaceae, Prevotellaceae, Enterobacteriaceae, Acidaminococcaceae,*

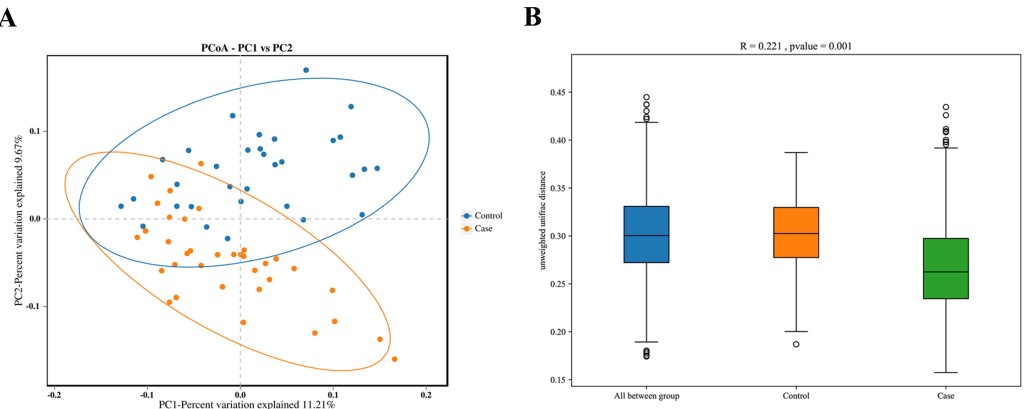

**Figure 2** **Comparison of beta diversity between the case and control groups.** (A) Principal coordinate analysis (PCoA) of unweighted Unifrac distances between the two groups; dots represent samples; (B) the groups indicated significant differences in similarity as examined by Anosim. Case, coronary atherosclerosis group; Control, healthy control group.

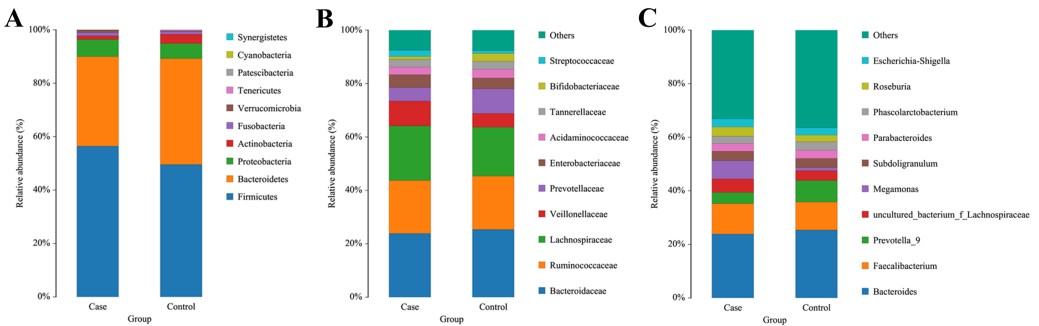

**Figure 3** **Dominant microbiota in stool samples.** The cylindrical accumulation map at the (A) phylum, (B) family, and (C) genus level. The relative abundance of the 10 most abundant bacteria in participants. Case, coronary atherosclerosis group; Control, healthy control group.

*Tannerellaceae*, *Bifidobacteriaceae*, and *Streptococcaceae* (Fig. 3B). There were no statistically significant differences in these top 10 families between the two groups.

At the genus level, the top 10 genera were: *Bacteroides*, *Faecalibacterium*, *Prevotella_9*, *uncultured_bacterium_f_Lachnospiraceae*, *Megamonas*, *Subdoligranulum*, *Parabacteroides*, *Phascolarctobacterium*, *Roseburia*, and *Escherichia-Shigella* (Fig. 3C). Among the top 10 genera, the only genus that differed significantly between the two groups was *Megamonas*, which was more abundant in the coronary atherosclerosis group (6.45% *vs* 1.10%, $P = 0.042$).

## Differences in gut microbial taxa

The microbial communities that contributed most to the differences between the two groups were examined by LEfSe analysis. The taxa with a *P* value <0.05 and absolute LDA (log10) scores >2 were the only ones considered significant. Eighteen bacterial

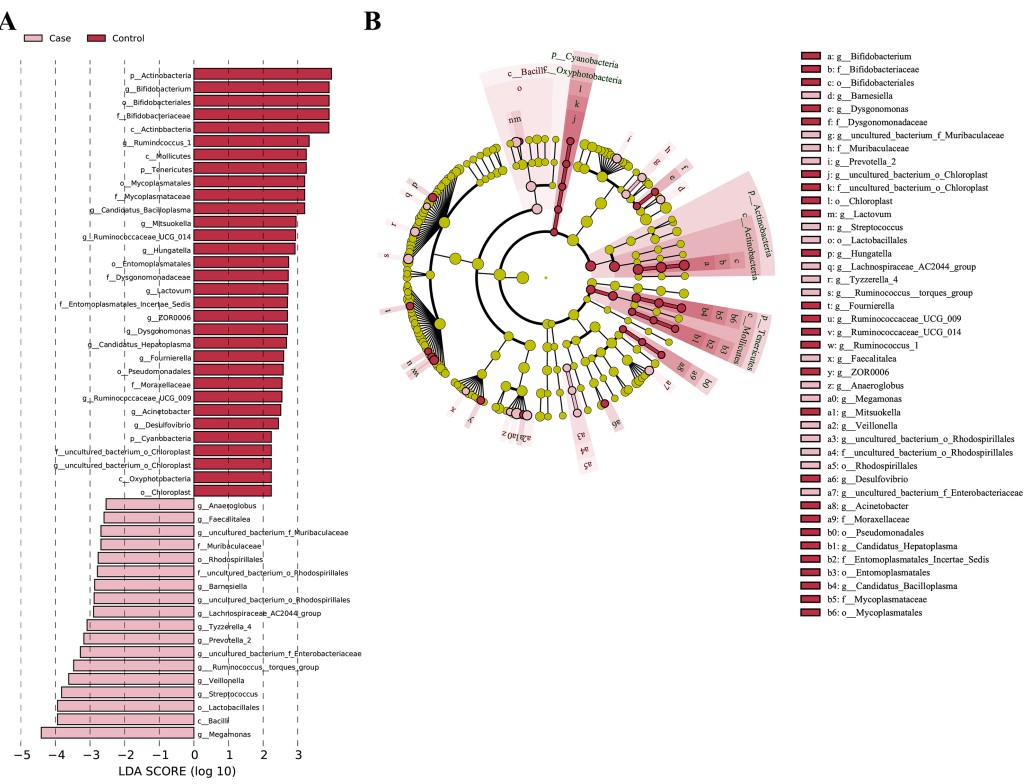

**Figure 4** **Taxonomic differences between the case and control groups.** (A) Differences in the gut microbiota in the case and control groups were tested by LEfSe analysis; (B) the phylogenetic distribution of microbial lineages associated with the two groups, indicated by cladogram. The levels represent phylum, class, order, family, and genus from the inner to outer rings. Case, coronary atherosclerosis group; Control, healthy control group.

species were enriched in the fecal samples of the coronary atherosclerosis group, and 32 were enriched in the control group. The abundance of several genera was notably higher in the coronary atherosclerosis group, including: *Megamonas*, *Streptococcus*, *Veillonella*, *Ruminococcus_torques_group*, *g_uncultured_bacterium_f_Enterobacteriaceae*, *Prevotella_2*, *Tyzzerella_4*, *Lachnospiraceae_AC2044_group*, *g_uncultured_bacterium_o_Rhodospirillales*, *Barnesiella*, *g_uncultured_bacterium_f_Muribaculaceae*, *Faecalitalea*, and *Anaeroglobus*. The genera that were abundant in the control group were: *Bifidobacterium*, *Ruminococcus_1*, *Candidatus_Bacilloplasma*, *Mitsuokella*, *Ruminococcaceae_UCG_014*, *Hungatella*, *Lactovum*, *ZOR0006*, *Dysgonomonas*, *Candidatus_Hepatoplasma*, *Fournierella*, *Ruminococcaceae_UCG_009*, *Acinetobacter*, *Desulfovibrio*, and *g_uncultured_bacterium_o_Chloroplast* (Fig. 4).

## DISCUSSION

Gut microbiota play a significant role in human physiology and pathology (*Clemente et al., 2012*). For instance, gut microbiota are involved in securing nutrients and metabolizing undigested food, as well as boosting the immune system and inhibiting inflammation (*Al*

*Bander et al., 2020*; *Chow et al., 2010*; *Gill et al., 2006*). An imbalance in the composition of the gut microbial community, known as dysbiosis, is a potential driver of heart disease, through what has been termed the 'gut-heart axis.' Dysbiosis in the gut microbiome may contribute to coronary atherosclerosis by triggering systemic inflammation, altering metabolite production, and translocating from the gut to plaque (*Tousoulis et al., 2022*). For this study, high-throughput technologies were applied to uncover a detailed composition of the gut microbiota associated with coronary atherosclerosis.

*Firmicutes* and *Bacteroidetes* are the dominant phyla of the healthy gut microbiome. Some changes occur to the abundance of these phyla, mainly in the form of an elevated *Firmicutes*-to-*Bacteroidetes* ratio, in some conditions of metabolic dysregulation, which may be because of the role *Firmicutes* plays in energy metabolism and calorie absorption (*Di Pierro, 2021*; *Indiani et al., 2018*; *Rinninella et al., 2019*). *Szabo et al. (2021)* found that the *Firmicutes*-to-*Bacteroidetes* ratio was greater in people with high carotid intima-media thickness. An animal study found an association between the size of atherosclerotic lesions and the *Firmicutes*-to-*Bacteroidetes* ratio (*Wang et al., 2021*). Likewise, our study found that coronary atherosclerosis was accompanied by an increase in the abundance of *Firmicutes* and a decrease in the abundance of *Bacteroidetes*, in agreement with the majority of existing studies (*Ramirez-Macias et al., 2022*; *Szabo et al., 2021*; *Wang et al., 2021*).

Comparing gut microbiota at the genus level between the two groups, we found that *Megamonas*, *Streptococcus*, *Veillonella* were significantly more abundant in the coronary atherosclerosis group than in the control group. Short-chain fatty acids (SCFAs) are crucial to the wide-ranging regulatory role of gut microbiota. The main SCFAs are acetate, propionate, and butyrate. *Perry et al. (2016)* suggested that the increased acetate production that occurs when the gut microbiota are exposed to calorically dense nutrients may increase ghrelin and insulin secretion, promote hyperphagia and energy storage as fat, and thereby promote obesity, hyperlipidemia, fatty liver disease, and insulin resistance. *Tirosh et al. (2019)* found that propionate in the diet was related to insulin resistance and obesity. Individual SCFAs affect atherosclerosis differently (*Yao, Chen & Xu, 2022*). Acetate acts as a substrate for cholesterol and promotes cholesterol synthesis. Conversely, propionate alleviates atherosclerosis by reducing vascular inflammation and lipogenesis (*Conterno et al., 2011*). The metabolic products of *Megamonas* in the intestine are acetate and propionate. *Megamonas* abundance is correlated with obesity, and is considered to be characteristic of Asian populations (*Kieler et al., 2017*; *Yachida et al., 2019*). In this study, there was a significant increase in the relative abundance of *Megamonas* found in the coronary atherosclerosis group compared to controls, and *Megamonas* was the key biomarker species screened by the LEfSe analysis. In addition to *Megamonas*, *Veillonella* is capable of fermenting amino acids, and many amino acids can become SCFAs (*Dai, Wu & Zhu, 2011*; *Neis, Dejong & Rensen, 2015*). An increased abundance of *Veillonella* has been associated with obesity and obesity-related inflammation (*Aranaz et al., 2021*), insulin resistance (*Moreno-Indias et al., 2016*), and nonalcoholic fatty liver disease (*Zhang et al., 2019*). *Zhang et al. (2019)* reported that the abundance of *Veillonella* was higher in patients with coronary atherosclerotic heart disease than in healthy controls. Additionally, *Veillonella* was found to be associated with coronary heart disease and identified as a commensal

bacteria in atherosclerotic plaque and fecal samples (*Chen et al., 2021*). *Streptococcus*, like *Veillonella*, was also found in a majority of atherosclerotic plaque samples (*Koren et al., 2011*), and the abundance of *Streptococcus* increased in the gut microbiome of individuals with atherosclerotic cardiovascular disease (*Jie et al., 2017*). Consistently, our coronary atherosclerosis group showed a significant increase in the relative abundance of *Veillonella* and *Streptococcus.* These results suggest that the microbial species with an increased abundance in the atherosclerosis group affect plaque formation in atherosclerosis, likely through the production of SCFAs.

Our study also found that *Ruminococcus_1* was less abundant in the coronary atherosclerosis group compared with the control group. Butyrate-producing species in the intestine are predominantly found in the *Firmicutes* phylum, and most specifically the *Ruminococcaceae* family. *Ruminococcus_1*, belonging to the *Ruminococcaceae* family, is one kind of butyrate-producing bacteria. Butyrate could lower the blood lipid level and up-regulate occludin expression, resulting in reduced inflammation (*Mao et al., 2019*). Inflammation is the common basis for physiological and pathological changes during the initiation and progression of atherosclerosis (*Libby, Ridker & Hansson, 2011*; *Ross, 1999*). A healthy plasma cholesterol level is also vital to cardiovascular health (*Ference et al., 2018*). *Ruminococcus_1* is involved in converting cholesterol to bile acids, and could help decrease cholesterol in the serum (*Huang et al., 2019*).

*Bifidobacterium* is an important intestinal probiotic that colonizes the intestinal mucus layer and is associated with health benefits in diseases such as type 2 diabetes (*Gurung et al., 2020*) and inflammatory bowel disease (*Coqueiro et al., 2019*). *Bifidobacterium* strengthens the intestinal mucus layer through autophagy and calcium signaling pathways, which is the mechanism behind the health benefits associated with it (*Engevik et al., 2019*). *Bifidobacterium* also plays a protective role in maintaining gut barrier functions and reducing systemic inflammation, and is thereby able to reduce the incidence of atherosclerotic cardiovascular disease (*Van den Munckhof et al., 2018*). Excessive plasma trimethylamine-N-oxide (TMAO), an intestinal metabolite, promotes the development of atherosclerosis (*Gregory et al., 2015*). *Bifidobacterium* can reduce plasma TMAO, which may also be a way it is able to reduce the risk of atherosclerosis (*Wang et al., 2022*). In line with these studies, our LEfSe analysis indicated that bacteria of five discriminant clades (phylum *Actinobacteria*, class *Actinobacteria*, order *Bifidobacteriales*, family *Bifidobacteriaceae,* and genus *Bifidobacterium*) were enriched in the control group. In summary, compared to the control group, most of the significantly reduced bacteria in the atherosclerosis group were probiotic bacteria in the intestine, which are essential to maintaining the intestinal barrier and to reducing systematic inflammation. The decreased abundance of these probiotic bacteria is accompanied by a series of preclinical and pathophysiological changes of atherosclerosis.

There are some limitations of this study. First, the sample size of this study is small, so the findings of this study should be validated in a larger sample. In addition, many risk factors, such as lipid levels, were not measured and so the associations between risk factors and intestinal bacteria were also not analyzed.

## CONCLUSIONS

Our study compared the alpha diversity, beta diversity, and microbial composition of the gut microbiome of adults with coronary atherosclerosis and healthy controls by measuring 16S rDNA from fecal samples. Alpha diversity did not differ between the coronary atherosclerosis group and the control group. Intergroup variation in community structure by beta diversity analysis demonstrated a separation between the two groups. At the genus classification level, the coronary atherosclerosis group exhibited a predominance of *Megamonas*, *Streptococcus*, *Veillonella*, *Ruminococcus_torques_group*, and *Prevotella_2*, *Tyzzerella_4*, while the abundance of *Bifidobacterium*, *Ruminococcus_1*, and *Candidatus_Bacilloplasma* was significantly lower in adults with coronary atherosclerosis. Our study demonstrated that adults with coronary atherosclerosis have characteristic gut microbiota, extending our knowledge of the association between gut microbiota and coronary atherosclerosis in China. These findings provide a theoretical basis for further study on the pathological mechanisms of coronary atherosclerosis, which could lead to individualized, targeted, intestinal microbial therapy with the goal of rehabilitating coronary atherosclerosis patients.

## ACKNOWLEDGEMENTS

We are grateful to Weitao Shen and Yue Li for their advice and help.

### Funding

The authors received no funding for this work.

### Competing Interests

The authors declare there are no competing interests.

### Author Contributions

- Yu Dong and Rui Xu conceived and designed the experiments, performed the experiments, analyzed the data, prepared figures and/or tables, authored or reviewed drafts of the article, and approved the final draft.
- Xiaowei Chen conceived and designed the experiments, prepared figures and/or tables, and approved the final draft.
- Chuanli Yang, Fei Jiang, Yan Shen, Qiong Li and Fujin Fang analyzed the data, prepared figures and/or tables, and approved the final draft.
- Yongjun Li performed the experiments, authored or reviewed drafts of the article, and approved the final draft.
- Xiaobing Shen conceived and designed the experiments, authored or reviewed drafts of the article, and approved the final draft.
## Human Ethics

The following information was supplied relating to ethical approvals (i.e., approving body and any reference numbers):

The Clinical Research Ethical Committee of Zhongda Hospital Affiliated to Southeast University granted Ethical approval to carry out the study within its facilities (Grant No. 2021ZDSYLL147-P01).

## DNA Deposition

The following information was supplied regarding the deposition of DNA sequences:

The 16S rDNA sequences described here are accessible *via* SRA accession numbers SAMN30930040 to SAMN30930106.

Sequences are available for review at https://doi.org/10.6084/m9.figshare.20936605.v3.

## Data Availability

The data is available at NCBI: SAMN30930040 to SAMN30930106; PRJNA881485.

## Supplemental Information

Supplemental information for this article can be found online at http://dx.doi.org/10.7717/peerj.15245#supplemental-information.

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
