# Peer review of "Characterization of gut microbiota in adults with coronary atherosclerosis"

_PeerJ, doi:10.7717/peerj.15245_

## Round 0.1 · original submission · Major Revisions

As the authors realize, the reviewers have found multiple issues regarding the design of the study as well as the findings presented in the manuscript. I therefore would like to invite the authors to respond to the reviewers' comments and implement them where possible.

Reviewer 1 ·

Basic reporting

The manuscript by Dong et al., analyzed the microbiota of patients with and without coronary atherosclerosis and they found some differences of gut microbiota between adults with coronary atherosclerosis and healthy adults.While the result is interesting, I only have a few comments list below. 

1, Line82-84. The authors collected samples from patients with and without coronary atherosclerosis. The information about age and gender may be useful if they have.  

2, Line89-91. The authors used the V3-V4 region for classification and degenerate primer 806R (5’-GGACTACHVGGGTWTCTAAT-3’. Can they provide references for this primer design?

 3, Line94-95. The amplified products were purified with agarose gel. Why was this step used? Because unspecific amplification? Can this introduce bias? 

4,Figure 2A. The two groups were overall separated in PCoA analysis, while there are some mixtures in the 2-D plot. Relate to comment-1, can the gender and age also have contributions in the PcoA plot, if they add gender and age group information. 

5, The author should give more detail about library preparations, such as how many cycles were used for adding adaptors and index, what kind of kit/sequencing were used.

Experimental design

.

Validity of the findings

OK.

Reviewer 2 ·

Basic reporting

The article is basically clear and unambiguous. The citation of the literature is not in place. Every point of view you said needs to be demonstrated. The article is a little less structured. The article structure is basically no problem. The data display is a little short of time. Some calculation formulas such as α β Shannon are added to make some descriptions, so as to enhance the sense of detail of the article.

Experimental design

There is no problem with the research design. The research question is to explain the differences, which is actually not meaningful.
Let me take a step back and say that there is no in-depth study of the differences you have studied. For example, you can improve the aspects of inflammation you are talking about, and carry out PCR verification of a key gene or protein in a mouse model. , which is slightly more argumentative.

Validity of the findings

The degree of novelty is basically not to mention, but it is quite satisfactory and the structure is the same
The original data provides more detailed,
The specific improvement content I also wrote in the pdf.

Additional comments

NO

Annotated reviews are not available for download in order to protect the identity of reviewers who chose to remain anonymous.

Reviewer 3 ·

Basic reporting

Authors use amplicon based metagenomics for identifying bacterial diversity in coronary atherosclerosis.
Details of the sequence data are missing, number of reads? Coverage?
Have any bacteria ever been linked with atherosclerosis before? Any such study conducted before or authors are first to report this? This lacking should be addressed in both introduction and discussion.
I found several studies on coronary artery disease and microbiota as well as atherosclerosis and microbiota. Authors should cite them and use material for introduction and discussion from them. DOIs of some are given for information.
10.1016/j.ebiom.2020.102649
10.3390/ijerph18084242
10.1161/JAHA.122.026036
10.1186/s12967-020-02539-x
10.3389/fcvm.2021.668532
An important study by Tuomisto et al. Age-dependent association of gut bacteria with coronary atherosclerosis: Tampere Sudden Death Study is missed. It should be incorporated.
Linkage of bacterial diversity with plaque should be identified if possible, as many relevant studies comment on that.

Experimental design

No references in this section of methodology at all. Authors should detail their analysis and provide references at all appropriate places as well as with softwares.
Do authors consider only chimeras to be contamination? Any explanation for contamination other than this?

Screening for differential colonies by Welch’s t test showed that at the genus level, in coronary atherosclerosis group, the microbial abundance was significantly increased in six genera. Please be informed that genus and genera are different things. How did authors screen colonies in statistical tests? Please explain?

It seems that authors do not have knowledge of microbiology and they thought this is easy way of making a paper, get samples, sequencing done commercially and analysis in cloud. Kindly take onboard someone who is well versed in these type of things and consult someone for guidance who has already published on this theme.

Validity of the findings

The abundance of several genera was noted. Is this genera : uncultured_bacterium_f_?
Are genera represented like this: Prevotella_2, Tyzzerella_4,?? Please make corrections.
Are you sure that your results are okay because Candidatus bacilloplasma is a novel lineage of Mollicutes associated with the hindgut wall of the terrestrial isopod Porcellio scaber and gut of shrimps? Has it been noted previously in elderly people of the age of your sampling cohort?
Kindly write the names of bacteria correctly and give links to literature for all these genus/specie etc.
On what basis the software is giving uncultured chloroplast signal? The amplicons were just for specie identification using specific regions? If a primer was not used for amplification of these fragments, how are they being picked by software?Is it usual to have this sort of thing because bacteria do not have chloroplast?

Discussion: This line should be edited or removed for clarity ‘As research on the pathogenesis of coronary atherosclerosis has progressed, traditional study directions show its limitations, leading us to turn attention to other factors that may be associated with the development of atherosclerosis. ‘
Veillonella is capable of fermenting amino acids, which can further converse into SCFAs? Please edit to clarify. English language mistakes rife throughout manuscript. Please get it edited for language and clarity by a professional scientific editor.
Ruminococcus_1, belonging to the Ruminococcaceae family? I have never come across _1 as a specie. Please edit.
Ruminococcus_1 involved in conversing of cholesterol to….I believe authors mean converting here.
p_Actinobacteria, c_Actinobacteria, o_Bifidobacteriales, f_Bifidobacteriaceae and g_Bifidobacterium ...What is p, c, o, f, g here?
Discussion should be improved and more reflective of what is going on in literature and link to the disease. Although few sentences on metabolite presence and their impact are there, it is not extensive or comprehensive at all. Conclusion section is missing and last line of discussion just states this provides basis for more study. Authors should improve this section as well.

Additional comments

Where was the raw data deposited? Accession number should be mentioned in data availability statement. Without it, the study should not be accepted as suspicions arise in case of missing data links due to paper mill business.

---

## Round 0.2 · Minor Revisions

As indicated by one of the reviewers, the manuscript still needs language editing by a native English speaker. Therefore, I would like to invite the authors to revise their manuscript accordingly for the last time.

Reviewer 1 ·

Basic reporting

The authors has adequately answered my questions. I am satisfied withe the authors' response.

Experimental design

none

Validity of the findings

The authors has adequately answered my questions. I am satisfied withe the authors' response.

Additional comments

none

Reviewer 3 ·

Basic reporting

The authors have now improved the manuscript in light of comments. however, language editing should be done carefully. e.g. no start of a sentence with And in abstract.

Experimental design

Well explained now

Validity of the findings

Valid

Additional comments

Accept after language editing

---

## Round 0.3 · accepted · Accept

The authors addressed the reviewers' comments during the revision rounds. The manuscript can now be accepted for publication. I congratulate the authors for their work.

Reviewer 3 ·

Basic reporting

Reviewed earlier
It is fine for acceptance now.

Experimental design

Ok

Validity of the findings

Valid